 eLIFE

# Plasmon resonance and the imaging of metal-impregnated neurons with the laser scanning confocal microscope

**Karen J Thompson[1†], Cynthia M Harley[2†‡], Grant M Barthel[3], Mark A Sanders[3], Karen A Mesce[2\*]**

[1]Department of Biology, Neuroscience Program, Agnes Scott College, Decatur, United States; [2]Department of Entomology, Graduate Program in Neuroscience, University of Minnesota, Saint Paul, United States; [3]University Imaging Centers Core Facility, University of Minnesota, Saint Paul, United States

**Abstract** The staining of neurons with silver began in the 1800s, but until now the great resolving power of the laser scanning confocal microscope has not been utilized to capture the in-focus and three-dimensional cytoarchitecture of metal-impregnated cells. Here, we demonstrate how spectral confocal microscopy, typically reserved for fluorescent imaging, can be used to visualize metal-labeled tissues. This imaging does not involve the reflectance of metal particles, but rather the excitation of silver (or gold) nanoparticles and their putative surface plasmon resonance. To induce such resonance, silver or gold particles were excited with visible-wavelength laser lines (561 or 640 nm), and the maximal emission signal was collected at a shorter wavelength (i.e., higher energy state). Because the surface plasmon resonances of noble metal nanoparticles offer a superior optical signal and do not photobleach, our novel protocol holds enormous promise of a rebirth and further development of silver- and gold-based cell labeling protocols.

**\*For correspondence:**
mesce001@umn.edu

[†]These authors contributed equally to this work

**Present address:** [‡]Department of Natural SciencesMetropolitan State University Saint PaulUnited States

**Competing interests:** The authors declare that no competing interests exist.

## Introduction

In the late 1880s, Santiago Ramón y Cajal began characterizing the cytoarchitecture of different types of brain neurons in a variety of animal species by employing the Golgi stain or 'black reaction' method. For reasons still unknown, only a random subset of neurons became labeled in their entirety, leading to our understanding that the nervous system is largely composed of individual cells separated by synapses. Although the Golgi method, which involves the deposition of silver, allowed for unprecedented insight into the morphological features of neurons, subsequent methods were developed so that specific neurons of interest could be characterized. For example, targeted neurons were filled with cobalt or nickel ions, and later intensified through the deposition of silver (*Pitman et al., 1972*; *Tyrer and Bell, 1974*; *Mesce et al., 1993a*). Eventually, the popularity of metal-filling and silver intensification methods gave way to the development and use of fluorescent compounds to label neurons (*Stretton and Kravitz, 1968*; *Stewart, 1981*; *Mesce et al., 1993b*; *Vitzthum et al., 1996)*. Furthermore, fluorescent specimens imaged with the laser scanning confocal microscope (LSCM), made available in the late 1980s (*Amos et al., 1987*), provided vastly improved results—in-focus 3D reconstructions of individual neurons were now easily obtained. With repeated laser imaging, however, a number of fluorophores in wide use showed susceptibility to photobleaching, thus lessening the ability of fluorescently labeled samples to be archived and repeatedly imaged over time.

The use of silver and gold particles for studies of cell and molecular biology, however, has had a recent resurgence because of the ability of metals to resolve small structural features in a variety of

**eLife digest** A fresh slice of brain tissue has a fairly uniform appearance, even when viewed under a microscope. To study the neurons and other cells in the brain, scientists must therefore first prepare tissue samples using methods that make it easier to see certain kinds of cells, or particular features of them. One method that has been available for over a century is to use metal particles to stain some of the cells. For example, when the Spanish anatomist Santiago Ramón y Cajal investigated how brain cells – or neurons – are organized in the brain in the late 1880s, he made the cells visible by staining them with silver. This silver staining technique, called the Golgi method, bears the name of Camillo Golgi who first discovered it. Both Golgi and Ramón y Cajal are considered to be the pioneers of neuroscience, and shared the Nobel Prize in Physiology or Medicine in 1906.

Over the years, silver staining was superseded by the use of fluorescent probes. Light travels in the form of waves, and different colors of light have different wavelengths (the distance between the peaks of the wave). Shining light of one specific color onto a fluorescent probe causes it to emit light of a longer wavelength. By detecting this emitted light, it is possible to visualize structures that contain the probes. In the late 1980s, the invention of the laser-scanning confocal microscope allowed highly detailed three-dimensional reconstructions of individual neurons to be obtained using these fluorescent labels.

Unfortunately, the lifespan of fluorescent probes is limited by the fact that their fluorescence decreases with repeated use, in a process called photobleaching. Traditional silver stains avoid this problem, but standard confocal microscopy cannot obtain good images from metal-stained cells. Now, Thompson, Harley et al. have overcome this problem by using the confocal microscope in a new way to detect emitted light with shorter wavelengths than the light that was initially absorbed (rather than the longer wavelength light normally detected). This protocol produced highly detailed three-dimensional images of individual metal-stained neurons that had been impregnated with silver or gold particles.

The short wavelength light is thought to result from the activity of free electrons called plasmons that are present on the surface of small metal particles (nanoparticles) that are about one millionth of a centimeter in size. When plasmons absorb radiation of a specific wavelength, they vibrate rapidly and emit their excess energy in the form of light. Medieval craftsmen unknowingly exploited this same phenomenon when they added silver and gold particles to molten stained glass, producing windows with vivid red and yellow colors that are still vibrant today.

A return to metal-based staining of brain tissue could produce similar longevity for today's tissue samples. Equally, the procedure developed by Thompson, Harley et al. opens up the possibility of revisiting archived material with the tools of modern confocal microscopy.

tissues. Enzyme metallographic technology, for example, enables the maximal sensitivity and resolution of cancer-related gene and protein expression patterns with either in situ hybridization or immunocytochemical techniques (*Downs-Kelly et al., 2005*; *Turzhitsky et al., 2014*). To date, these techniques and other modern methods have required conventional brightfield microscopy, involving the unwanted capture of out-of-focus light, and time-consuming reconstruction of serial histological sections or optical image planes.

It is clear that enabling the imaging of silver or gold particles with the LSCM would be of enormous benefit to neuroscientists and cell biologists alike. To this end, we discovered a novel strategy that permits the imaging of metal-impregnated cells with the LSCM. Silver- or gold-labeled neurons were imaged by using a method whereby the laser-induced photon emission, from any given sample, was captured at its higher energy state (shorter wavelength) as compared to the laser wavelength used to excite the metal-impregnated samples. This phenomenon can be used in the presence of high background fluorescence, and contrasts with conventional fluorescence microscopy whereby the emitted photons are of a longer wavelength (i.e. lower energy state). Our results are consistent with the hypothesis that the energy emitted after laser excitation is based on the phenomenon of silver (or gold) local surface plasmon resonance (*Willets and Van Duyne, 2007*) and not metallic reflectance or fluorescence.

Thus, the unexpected pairing of decades-old anatomical methods with newer confocal imaging technology is poised to unlock new information residing in a myriad of archived histological specimens. Furthermore, silver-impregnated preparations should retain their high-quality image for a century or more. For example, the original Golgi preparations created by Sanchez y Sanchez and Cajal in 1916 were recently beautifully reimaged using conventional brightfield microscopy (*Strausfeld, 2012*). We predict that an enormous reservoir of Golgi and other silver-stained nervous systems, from a wide range of vertebrate and invertebrate animals, can now be mined and easily imaged in their entirety. With the advent of enzyme metallography, these samples too can be visualized with unparalleled detail and reconstructed in three dimensions. Note, the imaging of silver-intensified cobalt-filled neurons with the LSCM was reported earlier in abstract form (*Thompson et al., 2011*; *2014*).

## Results

### Image quality across methods of microscope use

To appreciate the enhanced image quality produced by our new protocol, histological preparations of insect neurons (labeled with cobalt and intensified with silver) were imaged in multiple ways. First, a conventional brightfield image of a labeled neuron within a female abdominal ganglion of a grasshopper is presented. The cell body, neurites, axon and fine branching pattern can be seen (*Figure 1a*). However, this photomicrograph underscores the basic problem of brightfield microscopy when trying to illustrate the 3D structure of an entire neuron. First, the depth of focus is short relative to the size of the neuron. Second, the out-of-focus branches obscure the in-focus portions of the neuron. Thus, studies of cobalt-filled neurons are usually presented as camera Lucida tracings in which the cell's processes at all depths of focus are traced by hand onto paper (not shown); such drawings do not retain depth information. Stacked images from brightfield microscopy offer the promise of improvement (*Figure 1b*), but they too often lack the detail obtained using the LSCM.

We next imaged the same locust ganglion with the LSCM adjusted to the manufacturer's standard recommended wavelength settings. Such settings are typically based on laser excitation and emission wavelengths to detect fluorophores used in neuroanatomy, such as Alexa Fluor 488, Texas Red (TRITC), and Cy5. Based on these parameters, we observed strong autofluorescence from the ganglion while the silver-labeled neurons revealed a collective dark blur (*Figure 1c–e*).

Metals do, however, have the ability to reflect light. The reflectance mode of the LSCM has been utilized previously, especially in the diagnosis of skin cancers (*Calzavara-Pinton et al., 2008*). We thus examined our samples using this mode of operation whereby the excitation and emission wavelengths of the labeled neurons were relatively matched. Although some visualization of silver-impregnated neurons was observed, the reflective properties of the metal-impregnated neurons were poor, likely due to the thickness of the preparation and the refractive-index mismatch throughout the sample (*Figure 1f*).

### Imaging metal-impregnated neurons with the spectral LSCM

We employed a Nikon A1 Spectral Confocal Microscope to obtain all the confocal images presented here. The microscope was equipped with 4 standard photomultiplier tube (PMT) detectors and an array of 32 PMTs for spectral detection. When the settings of the LSCM were adjusted so that the laser excitation was of a longer wavelength compared to the shorter wavelengths selected for the detection of emitted photons, spectacular 3D electronic images of cobalt/silver impregnated neurons were produced; importantly, in-focus single optical planes could be stacked and rotated for three-dimensional renderings (*Figure 2a,b*). [Note: *Figure 2—figure supplement 1* shows details of post-capture image processing to remove the perineurial sheath. A video of the neurons rotating in 3D is also provided (*Video 1*).] Thus, silver-impregnated preparations imaged in this new way had all the positive attributes of fluorescently labeled preparations for which the confocal microscope has been so predominantly useful. Additionally, images could be obtained from older archived preparations; for example, the specimen illustrated in *Figure 3* was more than 25 years old. As fluorophores always emit light at a lower energy state (i.e. longer wavelength) than their excitation laser light, the settings we used (i.e. detection at shorter wavelengths) would not be applied during the standard operation of the LSCM.

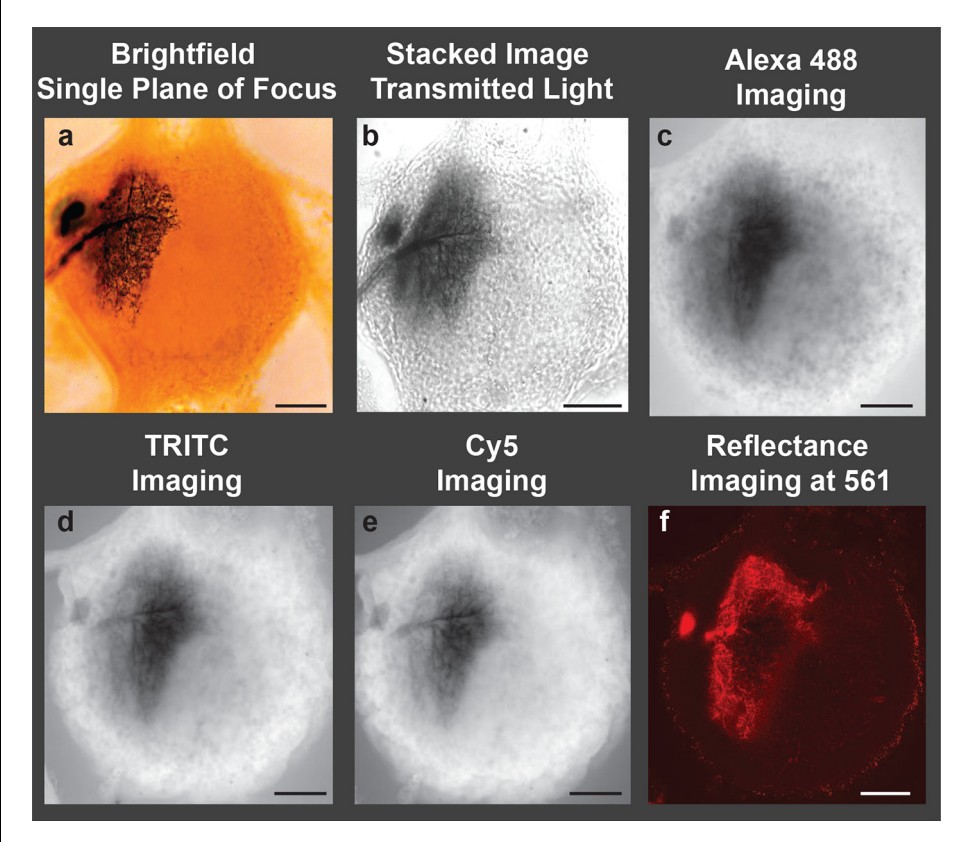

**Figure 1.** Image of a cobalt-filled, silver-intensified motoneuron, viewed with traditional microscopic imaging methods. View of an insect (grasshopper) ganglion containing a motoneuron labeled by retrograde transport of cobaltous chloride, which was placed in a well surrounding a transected lateral nerve. The cobalt-filled motoneuron subsequently received a deposition of silver using the Timm's silver intensification method *Tyrer and Bell, 1974*. All images presented here were taken using a 10× objective lens. (**a**) Brightfield image of the cobalt-filled silver-intensified motoneuon. (**b**) Stacked transmitted-light images acquired using the Nikon A1 spectral confocal microscope. (**c-e**) Confocal images obtained using appropriate settings for imaging specific fluorescent dyes; excitation and detection wavelengths were optimized for the specific fluorophore under consideration (listed in caption). (**c**) Alexa Fluor 488 setting: Sample was excited with a 40 mW argon multiline laser at 488 nm and emission was detected at 525/50 nm (center wavelength/bandwidth (FWHM)). (**d**) TRITC setting: Sample was excited with a 40 mW 561-nm diode laser and emission was detcted at 595/50 nm. (**e**) Cy5 setting: Sample was excited with a 10 mW 638 nm diode laser and emison was detected at 700/75 nm. (**f**) Imaging in the reflectance mode. Poor-quality images were obtained when the motoneuron was excited with a 561-nm laser line, and the emission was detected in a region over the same wavelength (using a notch filter). Only a portion of the neuronal arbors was visible and in focus. Scale bars = 100 μm.

## Optimization of excitation and detection wavelengths

How important is the specific laser excitation wavelength to achieving image quality? We addressed this question by exciting silver-impregnated samples at different wavelengths (405, 488, 561, or 638 nm; exact laser line indicated within the graphs) (*Figure 3*). We then measured the emissions over a three-decade range of wavelengths with a spectral detector (*Figure 3*). To compare the signal intensity across images and imaging types, we used a selection of four images from the scan and measured the emission intensity of both the region of interest (ROI) and the background (*Figure 3*). By taking a ratio of the intensity of the ROI and the background, we were able to calculate relative intensity and compare data across samples. In addition, all samples compared were acquired using a similar laser power setting (90% for the 561nm excitation, 100% for all other excitation wavelengths), pinhole size (1.2AU), and gain (120 for 561 nm, and 115 for all other excitations). Each of the

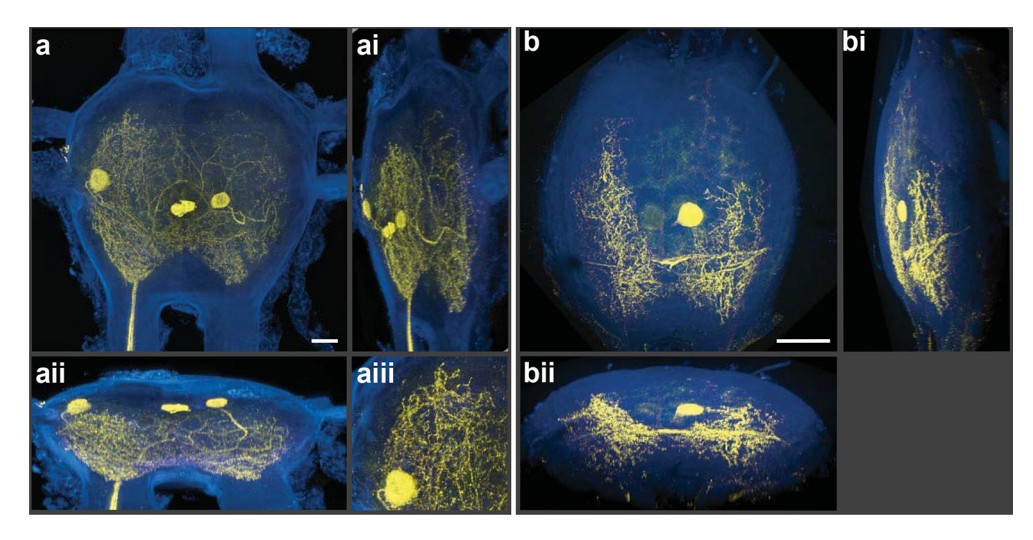

**Figure 2.** Silver-impregnated motoneurons imaged with the new protocol and the LSCM. Cobalt-backfilled and silver-intensified neurons, within two different insect ganglia, imaged with the LSCM. Images include selected regions of interest (i.e. somata and neural processes); signal from the ganglion's perineural sheath was removed (in FIJI). For additional information on image processing and the optical removal of the sheath, see the 'Materials and methods' and *Figure 2—figure supplement 1*. (a-aiii) For these images, a 561-nm excitation laser line was used and the photic emission was obtained from 490–530 nm. (ai, aii) Rotations of the neuronal image revealed the three-dimensional cytoarchitectual features of the neuronal arbors in both the lateral and dorsal-ventral planes. (aiii) A higher magnification of the left-side region of the ganglion (at 9 o'clock) highlights the level of detail observed among the fine branches visible using our new protocol. (b-bii) A different insect ganglion, showing a single cobalt-filled and silver intensified motoneuron. (b) Fine morphological details of an insect motoneuron and its ramifying branches are shown here. Laser excitation and emission collection were the same as in (a). Note that the neurite connecting the soma to its branches was not visible due to loss of tracer not image capture. (bi, bii) Lateral and dorsal-ventral rotations, respectively, demonstrating the ability of cells to be rendered in three-dimensions for analysis of arbor patterning. Scale bars = 100 µm. See video of neuronal rotations in the supplementary data section. LSCM: Laser scanning confocal microscope.

The following figure supplement is available for figure 2:

**Figure supplement 1.** Image processing of volume renderings and removal of perineurial sheath.

---

anatomical images shown (*Figure 3b–e*) is a collection of the entire spectrum of wavelengths to the left and right of the excitation wavelength.

We found that 561-nm and 640-nm excitation laser lines were required to produce strong emissions and high signal-to-noise images. The overall intensity of the shorter wavelength emissions with the 561-nm laser line was stronger (*Figure 3*), and hence, it became the laser setting of choice for the remainder of our studies. This is consistent with the peak absorption of silver and gold nanoparticles at ~550 nm (*Link and El-Sayed, 2003*).

Preparations were then stimulated with the 561-nm laser line to enable us to determine the optimal wavelength to use for detection of the emitted photons (*Figure 4*). We detected signal at 10 nm intervals along the range from 440 nm to 730 nm. Images represent groupings of 40 nm intervals. The images in the frames labeled 440–480 and 490–530 (*Figure 4c, ci*) showed a markedly stronger signal and low background compared to the other detection ranges (*Figure 4cii–v*). The graph at the bottom of the figure also showed the strong enhancement of the signal-to-noise ratio for its corresponding image that occurred in the 440–480 and 490–530 nm range of detection. It should be noted, however, that within the 440–480 nm range the high peak on the graph was not due to an abundance of signal, but rather the low noise (see graph inset). The second peak in the graph occurred where we had slightly higher noise but a much higher signal, resulting in greater imaging success when the signal was collected from 490–530 nm. With the short-wavelength

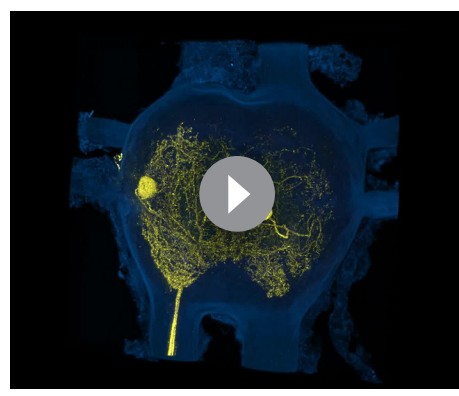

**Video 1.** Volume rendering of silver-impregnated locust motoneurons. Cobalt-filled and silver-intensified neurons within an insect ganglion. Image was processed as described in the 'Materials and methods'. Imaris software was used to generate a volume rendering and the video was exported as an AVI file. Handbrake software was used to convert the video to MP4 format. Video depicts same motoneurons as presented in *Figure 2a*.

detection setting of 490–530 nm, and a laser excitation setting of 561 nm, a bright neuron against a dark and low noise background emerged.

## Requirement for cobalt, silver, or gold

Can preparations that contain other metals than silver also be imaged with the LSCM? To examine this possibility, we looked at a number of histological preparations that had used different metals and impregnation techniques to characterize neuronal morphologies. Some neurons were: 1) filled with cobalt but had no silver intensification; 2) stained with silver but contained no cobalt (e.g. Golgi, reduced silver and Gallyas stains) (*Gallyas et al., 1980*); 3) stained with gold (Bodian's stain). Our findings indicate that silver or another noble metal such as gold was required. Cobalt alone produced poor-quality images, regardless of the excitation laser selected for use (*Figure 5*). Golgi stains of different brain regions from different species were also imaged, for example, the insect optic lobe (*Figure 5a*), mammalian pyramidal cell (*Figure 5b*), and mouse Purkinje cells (*Figure 5c*), all of which yielded high quality confocal images (excitation laser of 561 nm and detection of 430–740 nm). In addition, the Gallyas-stained sample (*Figure 5d*), the amino-cupric silver stained one (*Figure 5f*), and the gold-based Bodian's stain (*Figure 5e*) all produced quality neural images. The graph to the right of each set of images shows a plot that, essentially, indicates the signal-to-noise ratio for each sample. For images shown in *Figure 5a–f*, the best ratio in all samples occurred at 450 nm, and a secondary peak was located just above 500 nm. The insets of each plot show overall intensity of the signal in black and the background signal in gray. From these plots it is clear, however, that the high signal intensity just to the left of the excitation wavelength decreases to a lower value near 430 nm. Because of the lower signal in the 440–480 nm range, it is more reliable to detect the signal in the range of 490–530 nm.

In summary, the most straightforward and optimal way to implement the new methods communicated here is to image the metal-impregnated sample with the LSCM excitation laser set to 561 nm (or common 543 nm and 555 nm laser lines) and collect photons in the range of 490–530 nm.

## Discussion

In this study, we showed that the LSCM can be utilized to capture photons for image formation emitted by silver- or gold-impregnated neuronal specimens. Our analysis identified two constraints on imaging. One was that the microscope's longer wavelength laser lines (561 nm or 640 nm) were required for excitation. The second was that wavelength detection settings needed to be shorter, in the range of 440–530 nm. This relationship between stimulating wavelength and collecting wavelength contrasted with those involving fluorescent dyes. For example, fluorophores excited with a 561-nm laser line emit photons with wavelengths in the range of 570–610 nm, due to energy loss during the fluorescence energy transfer (Stokes Law). In fact, many LSCMs utilize filter assemblies that routinely filter out emission wavelengths shorter than the excitation laser line, thus preventing the type of capture we have outlined here.

The explanation, we believe, for both the presence of photic emissions from our silver and gold preparations and for the fact that they are of higher energy than the excitation wavelength, is the phenomenon of surface plasmon resonance. Plasmons are coherently oscillating free electrons on the surface of noble metal nanoparticles, such as silver and gold (*Willets and Van Duyne, 2007*). When the vector of incident light matches the oscillations of plasmons, there is an interaction or

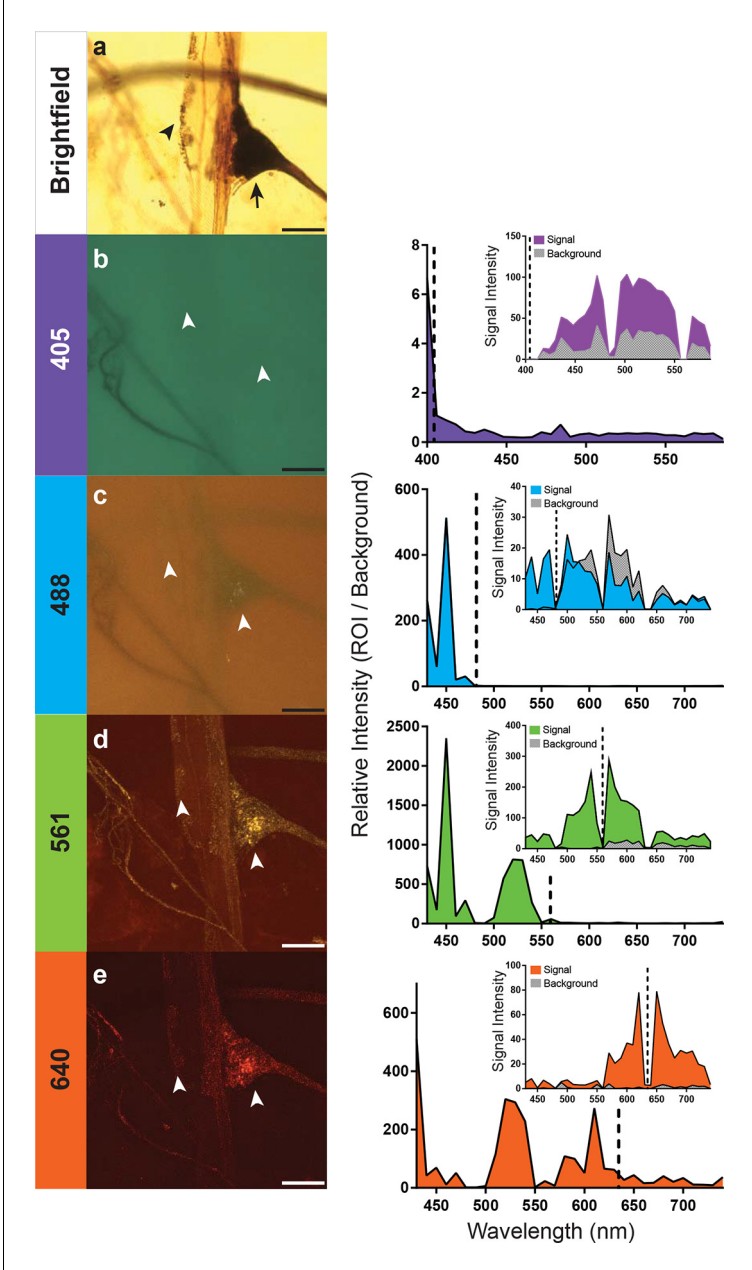

**Figure 3.** Optimal excitation and emission wavelengths to image silver-impregnated cells with the LSCM. Multiple images of a cobalt-filled silver-intensified stretch receptor organ, from a hawkmoth, were compared across different laser excitation wavelengths using the LSCM and compared to a brightfield image. (**a**) Standard brightfield image of the insect stretch receptor organ in the abdomen. The arrow points to the cell body of the sensory neuron, which is attached to a modified muscle fiber; the arrowhead points to efferent motor terminals that regulate the tension of the fiber. Other silver-impregnated fine nerves are out of focus in the background. (**b-e**) An excitation laser line of 405 nm (**b**), 488 nm (**c**), 561 nm (**d**), or 640 nm (**e**) was used to determine optimal wavelengths to obtain best signal-to-noise ratio images. Emissions are plotted to the right and color coded with neuronal image. With the 405 laser line, light was collected from 400 nm to 590 nm in 6 nm bins; this entire spectrum was used to generate the image depicted (**b**). The other images (**c-e**) were formed by collecting the spectrum from 430–740 nm (in 10 nm bins). The associated graphs indicate the signal-to-noise ratio calculated by determining the ratio of the intensity of emitted signal from a region of interest (ROI) divided by the intensity of the background. Insets within the graphs indicate the raw intensity values for the signal (black) and background (gray). In each trace the excitation laser wavelength is indicated as a vertical dashed line. Note that at the 561-nm
*Figure 3 continued on next page*

*Figure 3 continued*
laser excitation, the image quality is best. The nucleus of the stretch receptor is visible as are the efferent terminals on the muscle-like fiber; the previously out-of-focus fine nerve fibers are now in focus. Scale bars = 100 μm.

resonance, thus the term surface plasmon resonance (*Willets and Van Duyne, 2007*). Different sizes and shapes of plasmonic silver and gold nanoparticles, and their spacing, will produce different resonances that can vary in wavelength across the visible light range and the infrared. Photons can either be scattered in all directions by plasmonic nanoparticles or absorbed and transferred into heat or luminescence (*Zhang et al., 2007*; *Rycenga et al., 2011*). Of the noble metals and copper, known

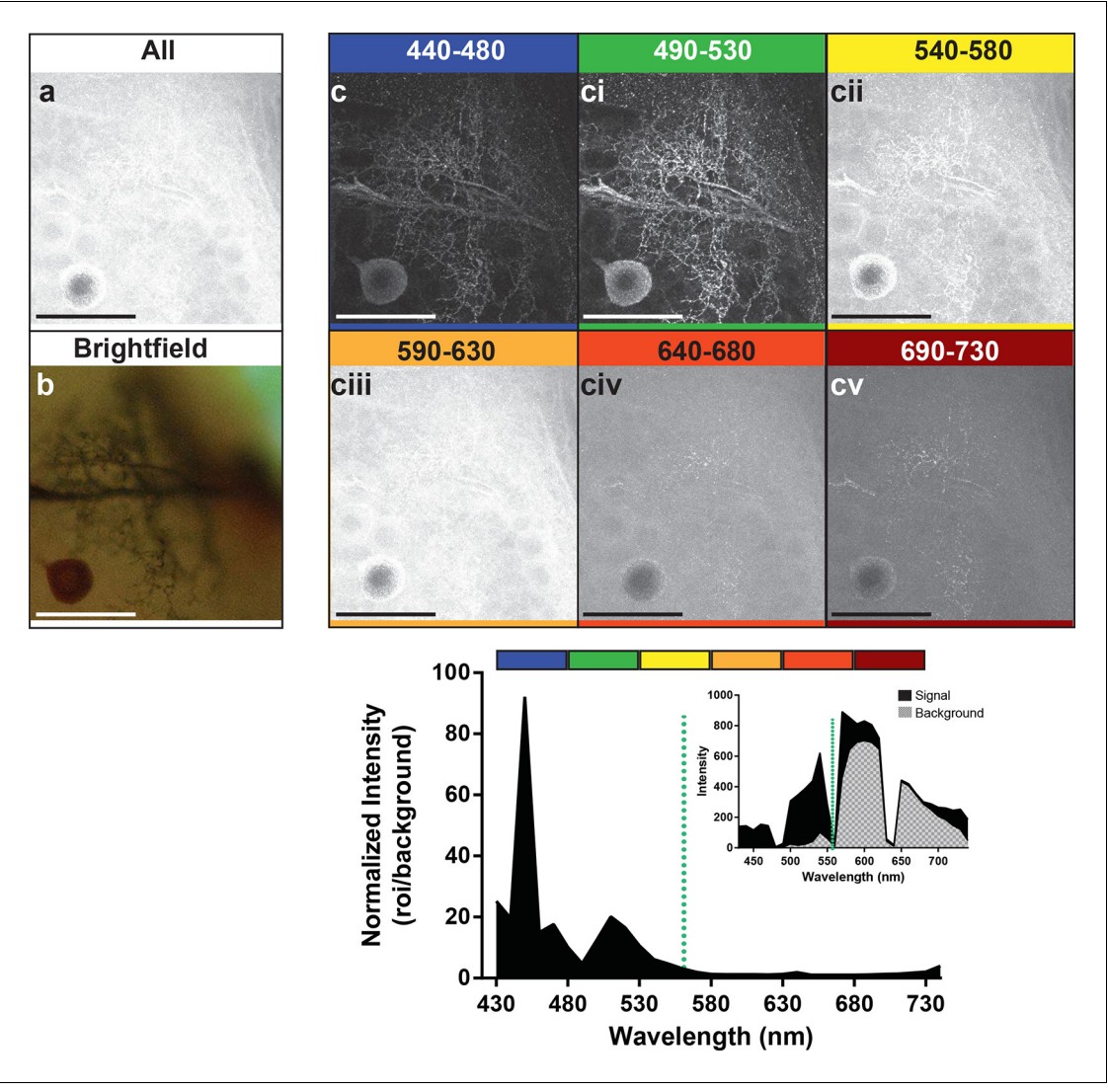

**Figure 4.** Optimal collection wavelengths to image silver-impregnated cells with the LSCM. Cobalt-filled and silver-intensified single insect motoneuron was imaged over a variety of different emission wavelengths (i.e. that which was collected). (**a**) Insect ganglion preparation imaged with a 561-nm excitation laser line and photic output collected over the full spectral range (430–750 nm). Note that only the soma is faintly visible. (**b**) Same cell imaged under standard brightfield conditions, where the cell body and numerous fine branches were now visible, but deeper branches were out of focus. (**c-cv**) Collected emissions from the 561-nm excited samples were parsed into 40 nm bandwidths. Clearly, the optimal ranges for signal maximization were in the shortest wavelengths, approximately 440–530 nm (see **c**). These same data are also shown graphically below the images, highlighting the strongly enhanced contrast of the signal at optimal wavelengths. It is noteworthy that at 440–480 nm the signal is relatively low (see inset), thus photic collection at 490–530 nm appears to be more optimal for image quality. Scale bars = 100 μm.

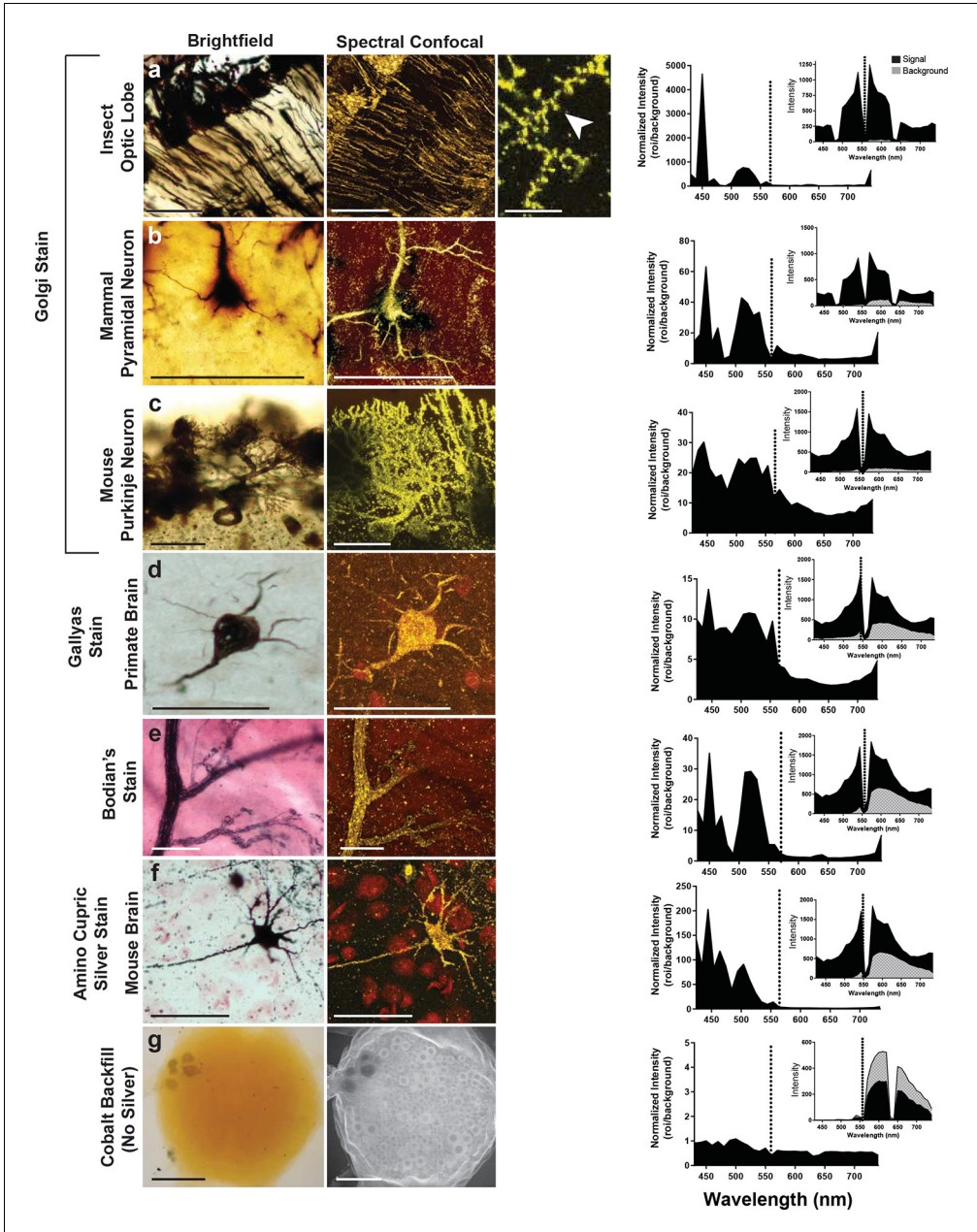

**Figure 5.** Examination of various metal-based stained neurons, across species, imaged with the LSCM. (a-f) Silver- and gold-impregnated tissues were imaged using standard brightfield microscopy (left) and compared to LSCM images excited with a 561-nm laser line (right). Normalized intensity graphs (signal to noise ratio) are shown to the right of each confocal image. Insets provide the raw intensity data for the detected signal (black) and detected background noise (gray). The dashed line indicates the wavelength of the excitation laser line that was used (561 nm). Preparations were as follows: (a) Golgi stain of a honeybee optic lobe. Note the presence of what appear to be dendritic spines (arrowhead); (b) mammalian pyramidal neuron; (c) mouse Purkinje neuron; (d) Gallyas stain of a monkey brain; (e) Bodian's stain of a mammalian motor end plate; (f) amino-cupric-silver stain of a mouse brain. In (g), insect neurons were backfilled with cobalt but not subsequently silver intensified. Without silver, the emission intensity profile is flat and the confocal image shows only faint arbor-free somata. Scale bars = 100 μm except for one bar in the far-right confocal image in (a) bar = 25 μm.

to exhibit plasmon resonance in the visible light range, silver is the most efficient for light capture. For example, a single nanoparticle can interact with 10 times more light than is physically incident upon it; thus, a silver nanoparticle is more efficient than any similarly sized chromophore (*Rycenga et al., 2011*; *Evanoff and Chumanov, 2005)*. Excitation and emission photons are not related in a one-for-one ratio, but rather, the excitation laser line (longer wavelength) provides the

energy input necessary to sustain the plasmonic light-emitting state of the metallic nanoparticles. The optical changes are large because of the strong interactions of light with noble metals (*Evanoff and Chumanov, 2005)*. While silver and gold are known to support surface plasmons at visible wavelengths, cobalt emits inefficiently in the 300 nm range—outside the collection wavelength setting available on the LSCM we used (*Kaminskienė et al., 2013*). Thus, the hypothesis that our silver- and gold-impregnated samples can be imaged as a consequence of surface plasmon resonance is supported by the fact that cobalt alone did not result in an emission in the visible light range (*Figure 5g*). Furthermore, the wavelength observed for silver and cobalt/silver nanoparticle plasmonic emissions is in the range of 440–530 nm (*Evanoff and Chumanov, 2005*), providing further support for our hypothesis.

Metallic nanoparticles of silver or gold have been exploited for use in medicine and biotechnology, in part, because they are nontoxic, do not photobleach, and are of relatively low cost (*Barnes et al., 2003*; *Iwasaki et al., 2006*; *Lal et al., 2007*; *Jain et al., 2008*; *Javier, 2008*; *Hamidi and Oskuei, 2014*; *Eustis and El-Sayed, 2006*). The use of plasmon-related technology is rapidly expanding in the biological fields and for other applications (*Downs-Kelly et al., 2005*; *Fan et al., 2014*). Metal nanoparticles are wavelength-tunable for enhanced optical contrast, they can be tagged with antibodies, and can serve as molecular sensors (*Turzhitsky et al., 2014*). One particularly interesting application involves cell labeling with nanoparticles coupled to fluorescent dyes (*Koyama and Tophyama, 2013*). Significant plasmonic enhancement of fluorescent dyes is observed, apparently the result of increased fluorophore stability permitting longer probing times with higher excitation energy; noble metal nanoparticle plasmons can enhance the signaling of fluorophores by two orders of magnitude or more creating exceptionally bright labels (*Jain et al., 2008*; *Demchenko, 2013*). It has also been shown that fluorophore-conjugated antibodies can be used to colabel antigens in cobalt-filled Timm's silver-intensified nervous system preparations (*Moos, 1993*), indicating that such preparations and others could be imaged with our new method and simultaneously with traditional ones appropriate for the emissions of the fluorescent antibody.

Silver emissions in our study revealed fine in-focus 1 μm or smaller fibers associated with insect neurons in wholemount (see *Figure 4ci*), which compared favorably to LSCM images of whole-mounted insect neurons labeled with the Cy-5 fluorophore, shown previously to yield a high signal-to-background ratio (*Mesce et al., 1993b*). In contrast to most fluorophores, however, silver and gold nanoparticles do not photobleach and have higher extinction coefficients—they can also be imaged with super-resolution microscopy methods and electron beams that surpass the ca. 200 nm diffraction resolution limit of a conventional LSCM (*Zhang et al., 2015*). Recently, local plasmon resonance and enhanced dark-field illumination, based on wavelength modulation, has resolved the coordinates of 80 nm silver nanoparticles with a 2.9 nm precision (*Zhang et al., 2015*). Because silver and gold nanoparticle plasmon resonance can generate a high signal to noise ratio (i.e. is bright), silver-labeled specimens like the ones we imaged have the potential to reveal structures at the nanometer scale when imaged with super-resolution microscopy methods. Although we have yet to determine the sizes and shapes of the silver nanoparticles deposited on the neurons we imaged, spectral and theoretical consideration should reveal such information (*Verbruggen et al., 2013*). By chemically controlling the sizes and shapes of silver or gold nanoparticles deposited on neuronal surfaces, the intensity and spectral characteristics of such cells could be customized and enhanced (*Evanoff and Chumanov, 2005*).

Other investigators have shown that nanoparticles of silver are themselves stabilized by the presence of cobalt. It has been noted that oxidation favors $CoO$ over $Ag_2O$, so that bimetallic nanoparticles with cobalt and silver in proximity would be expected to have reduced silver degradation, thus permitting more environmentally stable silver plasmonic behavior (*Saravanan et al., 2011*; *Sachan et al., 2012*). Note that cobalt was used to backfill the insect neurons in this study. The cobalt was first precipitated as $CoS$ that then served as the nucleus for silver deposition in the Timm's silver intensification procedure (see Methods). In traditional Golgi staining, black microcrystals of silver chromate form in stained cells, and these too have impressive longevity in excess of 90 years.

Our method, which does not rely upon fluorescent dyes, holds enormous potential for stimulating a reexamination of archived preparations, whether of Golgi-stained or cobalt-silver labeled nervous systems. By using the cobalt-silver intensification technique in combination with our LSCM protocols, new preparations can also be generated and imaged with ease, precision, and with great detail in

three dimensions. Such preparations are essentially permanent, and the information gathered from them increases the data available for characterizing neurons as individuals or as members of classes for comparative studies, adding to emerging neural information banks (*Parekh and Ascoli, 2013*). This archivability also promises to benefit clinical research and disease-related diagnostic techniques. Finally, just as plasmon resonance can explain the continued intensity of the red (use of silver nano-particles) and yellow (gold nanoparticles) colors found in centuries-old medieval stained glass and other works of art (*Sciau, 2012*), metal-impregnated neurons too will likely never fade, neither in their information content nor in their intrinsic beauty.

## Materials and methods

### Brightfield imaging
Brightfield images were taken using either a Nikon Eclipse E800 microscope, which allowed for extended depth of field capabilities using Nikon Elements (v. 4) software or with a Nikon Eclipse E400 that used ACT 1 software.

### Laser scanning confocal imaging
Confocal images were taken with a Nikon A1 spectral confocal microscope employing settings that achieved optimal Nyquist resolution. Even though there was no danger of photobleaching the sample, laser power was adjusted to avoid over saturation of the image during data acquisition. Spectral confocal imaging facilitates the capture of the up-converted signal, but it is not required for our new protocol as the plasmon resonance signal can be captured at wavelengths shorter than the 561-nm laser line excitation used in this study. The reflective and fluorescence default modes on a conventional non-spectral LSCM are suboptimal as they will catch only the tails of the peak of the signal. Although the excitation laser lines of 561 and 640 nm are standard on many LSCMs, the optical filter sets to capture light shorter than the excitation wavelength are not standard. If the LSCM were not tunable to detect up-conversion, a spectral detector or optical setup would need to be procured from the manufacturer of a given confocal microscope.

### Image processing and spectral unmixing for 3D rendering
A blind deconvolution algorithm in Fiji, an open source platform for image analysis (*Schindelin et al., 2012*), was used where indicated in figure legends to demonstrate further the ability of this method to show fine fibers. Additional spectral separation was performed using the Nikon Elements software on images where indicated within the figure legends. Where side views of confocal preparations were seen, these images were rotated using the software package Imaris. For the 3D rendering in *Figure 2*, image stacks were opened in Nikon Elements software and plasmon resonance channels were unmixed and the resulting channels selected and extracted from the spectral image. Additionally, one channel composed of autofluorescence (longer wavelength) from the ganglion was extracted to provide spatial context to the volume rendering. Each channel was exported in TIFF format and scaled to 8-bit depth. Images demonstrating plasmon resonance were opened in FIJI. Tools for 'oval', 'rectangle' or 'freehand' were used to select ROI in the images. The command 'fill' or 'clear outside' were then utilized to keep signal within the selected ROI or to discard signal outside of the ROI. Criteria for selection of a ROI included signal from the perineurial sheath as well as other signals that obscured motoneuron branches in 3D and orthogonal views. This method was done for all optical sections in the stack as well as for each extracted channel, autofluorescence channel excluded. All extracted channels were then merged into a single multichannel TIFF image. The resulting image was then pseudocolored in Elements. The autofluorescence channel and plasmon resonance channels were assigned unique color LUTS. For *Figure 2*, plasmon resonance channels were pseudocolored yellow and the autofluorescence channel was pseudocolored blue. Images were imported into Imaris software, and 3D renderings were created using the Surpass view. Orthogonal images were obtained using the 'Snapshot' feature. Animations were generated in Imaris by inserting a key frame and a corresponding 360 degree horizontal rotation (Mode = Maximum intensity projection, Rendering Quality = 1.000, Frame= no). Orthogonal images were saved in TIFF format and 3D animations were saved as AVI files. Some renderings were later converted to MP4 file format for presentation using Handbrake software.

Note: results can be achieved by blind unmixing spectral images and using the resulting channels for volume rendering and orthogonal images using Nikon Elements. 3D renderings can then be generated using Nikon Elements and/or FIJI software.

## Confocal image intensity measurements

To determine intensity values within confocal images, a selection of four images from the stack were used. A ROI with maximal labeling was chosen for measurements from four images as was a region within the sample background. A subset of four images was used in an attempt to keep intensity values comparable from sample to sample. In cases where the same preparation was imaged using different settings, an effort was made to make sure the same frames were used to obtain intensity values allowing for a more direct comparison.

## Tissue preparation

The labeled neuronal preparations shown in *Figures 1*, *2* and *4* were obtained from grasshopper (*Schistocerca americana*) neurons located in abdominal ganglia and were backfilled with cobalt chloride and processed with the Timm's silver intensification procedure (see references below for protocols and recipes). The stretch receptor organ shown in *Figure 3* was obtained from the hawkmoth, *Manduca sexta*, and filled with cobalt chloride and silver intensified as mentioned above. Additional preparations using other stains were loaned to us or purchased. NeuroScience Associates (Knoxville, TN), provided us with the amino cupric stain of mouse brain tissue (*Figure 5f*); the Golgi stain of mouse Purkinje cells, representing a standard protocol (*Ranjan and Mallick, 2010*) (*Figure 5c*); and the Gallyas stain of primate tissue, prepared according to previously described methods (*Braak and Braak, 1991*) (*Figure 5d*). The mammalian cerebral pyramidal neuron (*Figure 5b*) was a Golgi stained specimen purchased in 1969 from Ward's Sciences (West Henrietta, NY). The Bodian's gold-based stain (*Figure 5e*) of motor end plates was purchased from Carolina Biological, Burlington, NC. The insect optic lobe (*Figure 5a*) was stained with Golgi and provided by Dr. Susan Fahrbach (Wake Forest University) prepared according to methods described previously (*Farris et al., 2001*). We provided the cobalt-labeled slide (*Figure 5g*).

## Acknowledgements

Portions of this work were supported by grants from the National Science Foundation (IOS-0924155 to KAM), the University of Minnesota Agricultural Experiment Station (KAM), and the Agnes Scott College Gravatt Fund (KJT).

## Additional information

### Funding

| Funder | Grant reference number | Author |
|---|---|---|
| National Science Foundation | IOS-0924155 | Karen A Mesce |
| Agnes Scott College Gravatt Fund | | Karen J Thompson |
| University of Minnesota Agricultural Experiment Station | | Karen A Mesce |

The funders had no role in study design, data collection and interpretation, or the decision to submit the work for publication.

### Author contributions

KJT, CMH, KAM, Conception and design, Acquisition of data, Analysis and interpretation of data, Drafting or revising the article; GMB, Acquisition of data, Analysis and interpretation of data; MAS, Acquisition of data, Analysis and interpretation of data, Drafting or revising the article

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
