## [Decision Letter]

Thank you for submitting your work entitled "Plasmon Resonance and the Imaging of Metal-Impregnated Neurons with the Laser Scanning Confocal Microscope" for peer review at *eLife*. Your submission has been favorably evaluated by Eve Marder (Senior editor), and two reviewers, one of whom, Ronald L Calabrese, is a member of our Board of Reviewing Editors.

The reviewers have discussed the reviews with one another and the Reviewing editor has drafted this decision to help you prepare a revised submission.

Summary:

This is a technical paper that reports an advance in methods for reliable, easy three-D imaging of silver and gold impregnated neurons and other tissues. Using a LSCM, an instrument that is widely available, images from such impregnated neurons can be captured by collecting emissions at shorter wavelengths than used for excitation. Spectral analysis indicates that signal to background and image quality is maximized by excitation laser line of 561 nm and collecting emissions between 490-530 nm. Archived silver impregnated specimens are imaged at high quality. The observation that emission is largest at shorter wavelengths than excitation and that cobalt impregnated neurons are not imaged well while silver or gold ones are, suggests that the plasmon resonance associated emission underlies the imaging technique developed. The analysis appears carefully done and the data and images presented are convincing.

This is an exciting new technique and has wide applicability. One could potentially use silver stained tissues to see if finer details are revealed or perhaps connections that did not show up in the ordinary bright-field microscopes. Others may have their own uses and modifications of the technique. Historical archives of silver impregnated histological sections would also now be opened to easy three-D image capture and analysis.

A few issues should be clarified before publication.

1) The manuscript claims "...silver emissions offer greater spatial resolution and are more photostable than fluorophores...' but such a claim is not verified. While the images presented are beautiful and their stability without question, their resolution is not compared to similar high-quality fluorescence images. This comparison should be made and quantified.

2) Please amplify the supplemental methods section to more thoroughly describe post-capture image processing. For example, in Figure 2 the signal from the ganglion's perinerurial sheath was removed (in FIJI) and this process should be explained more fully and perhaps illustrated as a supplemental figure.

3) Please indicate whether the laser lines and detectors required are standard on most LSCMs.

---

## [Author Response]

*1) The manuscript claims "...silver emissions offer greater spatial resolution and are more photostable than fluorophores...' but such a claim is not verified. While the images presented are beautiful and their stability without question, their resolution is not compared to similar high-quality fluorescence images. This comparison should be made and quantified.*

We initially made the claim that silver emissions offer greater spatial resolution, because silver nanoparticle plasmon resonance is known to have a high signal-to-background ratio and extinction coefficient, and thus is more detectable (bright). This detectability enables small anatomical features to be more easily seen and imaged. In retrospect, we should have used the term “detectability” and not spatial resolution; thus we have modified our original assertion throughout the revised manuscript. Depending on objective lenses, light sources and mounting medium used, the diffraction limits of resolution for conventional microscopy, including the LSCM, are between 150-200 nm. The spectral LSCM we used cannot determine whether small neuronal fibers, for example, from a silver-stained preparation versus a fluorophore-labeled one are any better resolved below ca. 200 nm. Both silver- and fluorophore-labeling methods can offer high quality images using the LSCM as now stated in the text (paragraph four, Discussion). However, because the labeling we present involves metal particles, instead of less bright photo-bleachable fluorophores, such specimens can be probed with electron beams and newer super-resolution techniques that may well indicate that sub-cellular anatomical features can be better resolved using silver nanoparticles as compared to traditional fluorophores. Because such studies are beyond the scope of our report here, we have opted not to state that our image resolution is better than fluorescent labeling. In addition, we have added a section in the Discussion that presents the potential of super-resolution microscopy techniques for imaging silver or gold nanoparticles (paragraph four, Discussion).

2) Please amplify the supplemental methods section to more thoroughly describe post-capture image processing. For example, in Figure 2 the signal from the ganglion's perinerurial sheath was removed (in FIJI) and this process should be explained more fully and perhaps illustrated as a supplemental figure.

In response to the request for amplification on supplemental methods to more thoroughly describe post-capture imaging and the optical removal of the perineurial sheath covering the ganglion (for Figure 2), we have provided a new detailed section (in the Materials and Methods) with the heading, “Image processing and spectral unmixing for 3D rendering”. In addition to step-by-step procedures for post-capture imaging, we provide a new supplement figure (Figure 2—figure supplement 1), which illustrates the image processing from the raw state to the processed image.

*3) Please indicate whether the laser lines and detectors required are standard on most LSCMs.*

We further clarified that the laser lines and PMT detectors are standard on all commercially available spectral LSCMs. For non-spectral LSCMs, we mention that the excitation laser lines are standard, but that the optical filter sets to capture light shorter than the excitation wavelength are not standard. If the LSCM were not tunable, a spectral detector or optical setup would need to be procured from the manufacturer of a given confocal microscope (subheading “Laser scanning confocal imaging”).